# Distributed Event-Triggered Secondary Recovery Control for Islanded Microgrids

**Xiaofeng Wan ***, Ye Tian, Jingwan Wu , Xiaohua Ding and Huipeng Tu

School of Information Engineering, Nanchang University, Nanchang 330031, China;
411018719031@email.ncu.edu.cn (Y.T.); 401030820003@email.ncu.edu.cn (J.W.);
351032718001@email.ncu.edu.cn (X.D.); 411016620186@email.ncu.edu.cn (H.T.)
\* Correspondence: xfwan@ncu.edu.cn

**Abstract:** Distributed cooperative control methods are widely used in the islanded microgrid control system. To solve the deviation of frequency and voltage caused by the droop control, it is necessary to recovery the frequency and voltage to the rated value using a secondary control strategy. However, the traditional communication method relies on the continuous periodic one, which makes the communication burden of the islanded microgrid system heavy and conflicts with the actual operation of the power grid. Using the secondary recovery control method based on the distributed event-triggered method, we conserve communication resources by reducing the number of transmissions of sampled data and achieving the recovery control of the frequency and voltage and the original proportional sharing of active power. In addition, we analyze the stability of the distributed event-triggered strategy and build a microgrid system with MATLAB/Simulink to verify the effectiveness of the control method. Furthermore, we compare with a traditional periodic communication system and demonstrate the superiority of our distributed event-triggered approach.

**Keywords:** distributed control; multiagent system; interevent interval; consensus control; secondary control; droop control

## 1. Introduction

New models of power systems have started a centralized power supply with distributed generation (DG). The integration of DGs into the power grid presents significant risks to the safe and stable operation of the main power grid and the quality and reliability of the power grid cannot be guaranteed [1]. In order to solve these problems, Ali Bidram and Josep M. Guerrero proposed a microgrid system with the improvement of modern control technology and power electronic interface technology [2,3]. The complex control structure of the microgrid and the diversity of control objectives has driven [4,5] to propose a hierarchical control method for the system that is divided into three layers, namely primary control, secondary control, and tertiary control. In primary control, in order to solve the frequency and voltage deviation problems caused by droop control to the microgrid system, an effort must be made to eliminate or reduce these deviations during the secondary control. Zhou Qixun and Yang Zhichun proposed an improved droop control strategy that can reliably control bus voltage and frequency but cannot eliminate the frequency deviation [6,7]. WU Beibei and Alexander Micallef use a centralized control method to send a control signal to the primary controller, which can achieve voltage and frequency stability [8,9]. However, it has a high demand for the microgrid central controller and communication environment. Finally, the tertiary control enables a reasonable distribution of power among distributed sources by adjusting the secondary control's set value, as mentioned in [4,5,8]. This paper focuses on the secondary control, so the tertiary control will not be repeated.

In order to reduce the dependence on the microgrid controller and the communication network, Xin Huanhai proposes the use of a fully distributed method of hierarchical control

for frequency regulation and power optimization [10]. However, there is no communication link between each DG, and it is difficult to coordinate and control. Reza Olfati-Saber proposed and applied a distributed consistency control strategy to the microgrid [11,12]. DGs are considered agents of the system. Every agent utilizes the communication information of itself and neighboring agents to achieve autonomous operation. At the meantime, all agents work together and achieve control goals jointly to eliminate dependence on the central controller [13,14]. Reza Olfati-Saber summarizes the necessary and sufficient conditions for the communication topology to be asymptotically consistent for a strongly connected balanced graph system [11]. Chen Meng employs an internal model approach and a distributed secondary frequency and voltage control strategy to compensate for the frequency and voltage deviations to achieve a control effect [15]. Due to the uneven distribution of reactive power in traditional droop control, reactive power in [16] is redistributed through distributed algorithms, and voltage regulation is performed. In addition, the secondary control of the virtual synchronous generator also utilizes a distributed control method [17]. However, these methods are based on periodic sampling data control and the communication burden between DGs is still high [18] and uses a discrete-time control algorithm for the secondary control problem of island microgrids. In practical applications, the communication network is often limited. It is necessary to reduce the communication burden as much as possible. Reducing unnecessary communication between DGs and the frequency of DG communication will also extend the life of related sensors [19].

Recently, event-triggered control mechanisms have been widely used in multiagent distributed control. In [20], the mechanism of event triggering is discussed and the event triggering is applied to a first-order multiagent system. There is an event-triggered consensus problem for multipath systems with single integrator dynamics [21]. Zhang Hao investigates the event-triggered consensus problem for multiagents with general linear dynamics [22]. Li Zhenxing proposed a method of event-driven consistency tracking, which could achieve better synchronization with or without a central controller [23]. Yang Dapeng examines the decentralized events of linear multiagent systems for trigger consistency under generalized directed graphs [24]. Similarly, in the microgrid control both centralized and distributed control methods consider event-triggered communication [25]. Xiao Xiangning developed an event-triggered controller for the secondary voltage control in the microgrid island operation mode [26]. It used a state observer for maintaining the DG state to achieve precise voltage control, but did not consider the frequency recovery control.

In this paper, we examine the secondary frequency and voltage recovery control and power sharing of the microgrids with the communication burden, because analyzing the control strategy can be triggered by distributed events. The distributed control scheme is displayed in Figure 1. The non-periodic communication method differs from the traditional periodic control method in order to avoid the continuous exchange of information between the DGs and to reduce the demand for the communication network bandwidth. We design a corresponding trigger function to control the trigger time of DG. Through stability analysis, we use Lyapunov stability theory to demonstrate that the proposed control method can maintain the global stability of the multiagent system (MAS). Finally, we carried out simulation verification in MATLAB/Simulink and compared it with the traditional distributed method, verifying the effectiveness of the distributed event-triggered method.

The structure of this paper is as follows. Section 1 provides some backgrounds of islanded microgrid control, including hierarchical control, distributed control, and the application of distributed event-triggered control at this stage. Section 2 analyzes the primary control carefully to explain the reasons for adopting the secondary control. In Section 3, the distributed event-triggered controller is proposed and analyzed. Then in Section 4, the Lyapunov method is used to analyze the stability of the distributed event-triggered system. The simulation results are given in Section 5.1 and compared with the traditional distributed microgrid control in Section 5.2. Finally, conclusions are given in Section 6.

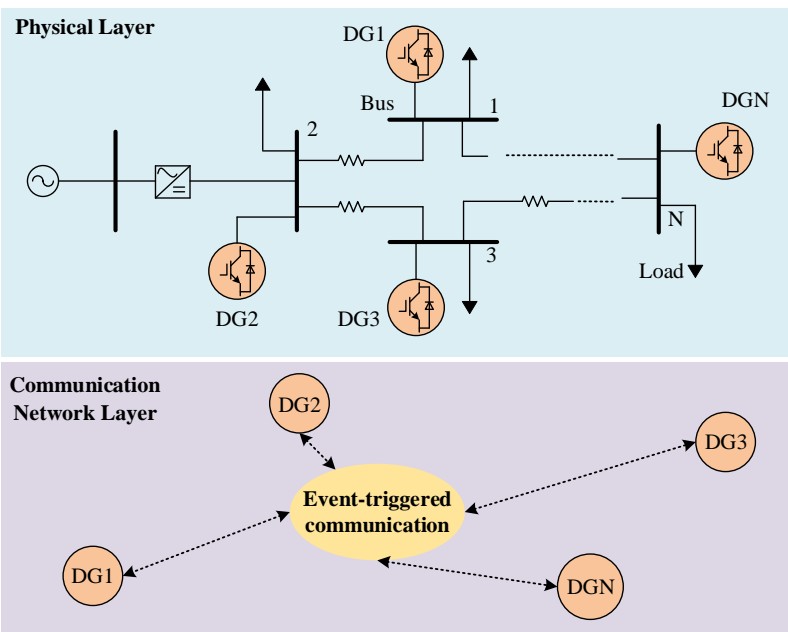

**Figure 1.** Distributed control system.

## 2. Primary Control of Islanded Microgrids

It is essential to employ primary control in the island mode to realize a balanced distribution of power based on the energy demand of DGs while maintaining voltage and frequency stability in the microgrid. Power electronic inverters are used because DGs are connected to microgrids. The switching devices of the inverters are the insulated gate bipolar translator (IGBT) and the control methods are usually pulse width modulation (PWM). The typical control strategy implemented by the inverter under the dq coordinate is a multiloop control setup that distributes power in the outer loop, while the inner loop consists of voltage and current double closed loop strategies. The power outer loop mainly generates the reference command for the inner loop to carry out the droop control function. The inner loop's double closed-loop control goes along with it to track the reference command value of the power outer loop to accomplish active and reactive power decoupling. The droop control method simulates the power frequency and static characteristics of synchronous generator sets. Due to the low inertia of the microgrid under inverter control, the power curve follows linear laws. Based on the inertia of the synchronous motor, $\omega_i$ and $v_i$ are adjustable by changing the power point on the droop curve, which is expressed as:

$$\omega_i = \omega_{ni} - m_i P_i \tag{1}$$

$$U_i = U_{ni} - n_i Q_i \tag{2}$$

The secondary control provides $\omega_{ni}$ and $U_{ni}$. Droop control adjusts $\omega_i$ and $v_i$ primarily to achieve a power balance. The main control allows DGs to automatically share active and reactive power. The load fluctuation of the microgrid can significantly affect the frequency, voltage, and stability of the microgrid. In this situation, the droop characteristic will cause the frequency and voltage values of the microgrid to stray from the rated values. Low-pass filters are generally used to obtain $P_i$ and $Q_i$ components of the inverter output power in order to remove the high frequency ones

$$P_i = \frac{\omega_c}{s + \omega_c} p_i \tag{3}$$

$$Q_i = \frac{\omega_c}{s + \omega_c} q_i \tag{4}$$

By transforming $abc/dq$, calculate $P_i$ and $Q_i$

$$P_i = v_{di}i_{di} + v_{qi}i_{qi} \tag{5}$$

$$q_i = v_{qi}i_{di} + v_{di}i_{qi} \tag{6}$$

Droop control is mainly used so that the output power of each inverter can be evenly distributed according to requirements. The link of power distribution is mainly achieved by adjusting the droop coefficients $m_i$ and $n_i$. Generally, angular frequency under rated conditions is chosen as $2\pi \times 50$ rad/s, namely, the set parameter meets $\omega_{n1} = \omega_{n2} = \ldots = \omega_{nn} = 100\pi$. The appropriate $m_i$ is selected so as to meet $m_1 P_1^* = m_2 P_2^* = \ldots = m_n P_n^* = K_1$, and the no-load angular frequency of each DG droop curve is then equal. Currently, the inverters are connected in parallel and the output frequencies of each DG are also equal. Then

$$m_1 P_1 = m_2 P_2 = \ldots = m_n P_n = C_1 \tag{7}$$

$$P_1 = \frac{C_1}{m_i} \tag{8}$$

Therefore, based on the satisfaction of (1), the inverter output power is inversely proportional to its droop coefficient. In general, the proportional distribution of inverter output power can be achieved by adjusting each of the droop coefficients, that is, by adjusting the slope of the droop curve. Similarly, set $U_{n1} = U_{n2} = \ldots = U_{nn} = 311v$ and select the appropriate $n_i$ such that:

$$n_1 Q_1 = n_2 Q_2 = \ldots = n_n Q_n = C_2 \tag{9}$$

$$Q_1 = \frac{C_2}{n_i} \tag{10}$$

considering the discussion above, in the primary control, the droop control is implemented to yield a proportional distribution of power. Due to the fact that droop control is a differential control, frequency deviation will occur. Therefore, in the secondary control, the frequency deviation should be further corrected.

## 3. Secondary Control of Islanded Microgrids

We treat each DG as an agent, the islanded microgrid system is a MAS. A DG only needs to stabilize its voltage and frequency according to the local control strategy, there is no mutual communication. Thus, they cannot coordinate with other DGs to complete system-level goals. In the secondary control, frequency, voltage, and power must be managed and optimized and the existence of a communication network in each DG must also be managed and optimized. The secondary control strategy in this paper avoids the need to collect global information. DGs achieve global information sharing through local information exchange between buses. With the limited capacity of communication, the amount of information exchanged between the DGs is relatively large and the event-triggered control method was adopted to reduce the sampling frequency of the system, thus enabling the system to meet the required control objectives and maintain accuracy. Control objectives can be expressed as follows:

$$\lim_{t\to\infty} (\omega_i(t) + \Delta\omega_i(t)) = \omega_{ref} \tag{11}$$

$$\lim_{t\to\infty} (V_i(t) + \Delta V_i(t)) = V_{ref} \tag{12}$$

### 3.1. Graph Theory

In a MAS, $G = \{V, \varepsilon, A\}$ describes the information interaction between agents. $V = \{v_1, v_2, \ldots, v_n\}$ represents the time-space collection of nodes, each node represents an

agent, and each DG is regarded as an agent. This is the collection of edges between node pairs, that is, $\varepsilon \in v \times v$. $A = (a_{ij}) \in R^{N \times N}$ is the weight matrix for an edge in $G$. $(v_i, v_j) \in \varepsilon$ means node $i$ and node $j$ are neighbors and the link communication function is established. $a_{ij}$ is the weight of edge $(v_i, v_j)$. When $(v_i, v_j)$ belongs to $\varepsilon$, $a_{ij}$ is positive and is usually 1, otherwise $a_{ij}$ equals 0. Generally, the node itself is not allowed to pass, therefore there is no self-loop edge. The degree matrix of the node $i$ is defined as $d(v_i) = \sum_{j=1} a_{ij}$. $L = D - A$ equals the Laplace matrices, where $D$ is the in-degree matrix of $G$. For convenience, the communication network of the islanded microgrid is also denoted as $G$.

### 3.2. Distributed Secondary Controller Design

DGs are connected by a small number of communication cables. When the load changes in the microgrid, each DG obtains the current state of itself and its neighbors. Then it calculates accordingly to achieve complete distributed control. Therefore, this can eliminate the need for centralized data centers and control.

#### 3.2.1. Frequency Recovery

Due to the power calculation error generated by the inverter and the measurement link itself and the power redistribution error generated by the distributed algorithm, $m_i P_i$ may no longer be equal after the load changes. In this way, each inverter calculates its own compensation amount separately, the frequency compensation amounts of the inverters in parallel can differ and cannot be coordinated.

For each DG output frequency $\omega_i$ to return to the set value $\omega_{ni}$, the frequency compensation required by the $i$th inverter is $m_i(P_i - P_i^*)$. Inverters are compensated based on the average of each DG compensation amount, namely:

$$\Delta \omega = \frac{1}{n} \sum_{i=1}^{n} \Delta \omega_i = \frac{1}{n} \sum_{i=1}^{n} \Delta m_i (P_i - P_i^*) \tag{13}$$

using $P_{i(p.u)} = P_i / P_i^*$ as the substitute, Equation (13) is further simplified to give:

$$\Delta \omega = m_1 P_1^* \left( \frac{1}{n} \sum_{i=1}^{n} P_{i(p.u)} - 1 \right) = K_\omega \left( \frac{1}{n} \sum_{i=1}^{n} P_{i(p.u)} - 1 \right) \tag{14}$$

In Equation (14), $K_\omega = m_1 P_1^*$, the average value of each DG active power per unit is $\sum_{i=1}^{n} P_{i(p.u)} / n$. The fast dynamics may be ignored and the state-space model of the active power per unit value can be constructed as follows:

$$\begin{cases} \dot{P}_{i(p.u)}(t) = u_{\omega i}(t) \\ y_{\omega i}(t) = P_{i(p.u)}(t) \end{cases} \tag{15}$$

the state variable is $P_{i(p.u)}$, in the case of a distributed controller:

$$u_{\omega i}(t) = \gamma \sum_{j \in N_i} a_{ij} \left( P_{j(p.u)}(t) - P_{i(p.u)}(t) \right) \tag{16}$$

$P_{i(p.u)}(t)$ is then substituted into Equation (14) to obtain the frequency compensation amount of DGi at time $t$:

$$\Delta \omega_i(t) = K_\omega \left( P_{i(p.u)}(t) - 1 \right) \tag{17}$$

final convergence value $P_{(p.u)\infty}$ represents the average value of the active power per unit value, from which the total compensation amount of the required frequency can be calculated, which is given by:

$$\Delta \omega = K_\omega \left( P_{(p.u)\infty} - 1 \right) \tag{18}$$

According to Equations (15) and (16), it can be seen that this distributed control method relies on the continuous state feedback, which means that communication between DGs is relatively heavy. The communication transmission pressure of sampling data can be effectively reduced in distributed control by an event triggering consistency control. In this paper, we propose a distributed event-triggered controller to reduce the communication between data generators and ensure the gradual stability of the system [24].

In the event-triggered distributed control law, we can redefine Equation (16) as follows:

$$u_{\omega i}(t) = \gamma \sum_{j \in N_i} a_{ij} \left( \hat{P}_{j(p.u)}(t) - \hat{P}_{i(p.u)}(t) \right) \tag{19}$$

The superscript represents the value of the corresponding variable at the moment of the trigger and is defined as:

$$\hat{P}_{i(p.u)}(t) = P_{i(p.u)} \left( t_k^{\omega i} \right), t \in \left[ t_k^{\omega i}, t_{k+1}^{\omega i} \right) \tag{20}$$

If DGi and $t \geq 0$, define the measurement error of $P_{i(p.u)}$ as follows:

$$e_{\omega i}(t) = P_{i(p.u)} \left( t_k^{\omega i} \right) - P_{i(p.u)}(t) \tag{21}$$

DGi trigger function is as follows:

$$f_i(t, e_{\omega i}(t)) = \| e_{\omega i}(t) \| - c_\omega e^{-\alpha_\omega t} \tag{22}$$

$c_\omega > 0$ and $\alpha_\omega$ are the normal numbers to be determined. The event trigger time $t_k^{\omega i}$ is defined as follows:

$$t_k^{\omega i} = \inf \left\{ t > t_{k-1}^{\omega i} \middle| f_i(t, e_{\omega i}(t)) = 0 \right\} \tag{23}$$

Figure 2 shows how the event trigger time is generated. With this control strategy, each DGi controller continuously monitors its own status. DGi generates a trigger event if $P_{i(p.u)}$ of DGi's state variable exceeds a certain threshold, $f_i(t, e_{\omega i}(t)) \geq 0$. DGi uses its current state as input to its controller and broadcasts its current state to its neighbors DGj. In addition, the measurement error of the state variable $P_{i(p.u)}$ of DGi is reset to zero.

**Theorem 1.** *Even if the matrix L does not change during the control process of the MAS, driven by the control law Equation (16) of each agent in the MAS, the system can gradually reach the same point if and only if the corresponding G contains a directed spanning tree [9,10].*

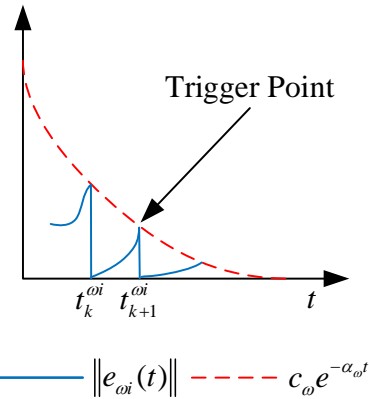

**Figure 2.** Trigger time.

In Equations (15) and (19), a distributed controller ensures the global stability of the active power per unit dynamic system. Combining (1), (15), (17) and (19) will give you the frequency setting value:

$$\omega_{ni}(t) = \int [\omega_i(t) + \Delta\omega_i(t)]dt. \tag{24}$$

### 3.2.2. Voltage Recovery

As (13), the output voltage $v_i$ of each DG returns to its set value $v_{ni}$, and the average of the voltage compensation of each inverter is used as the voltage compensation of each inverter, namely:

$$\Delta v = \frac{1}{n}\sum_{i=1}^{n} v_i(Q_i - Q_i^*) \tag{25}$$

by substituting in $Q_{i(p.u)} = Q_i / Q_i^*$ for (25) we get:

$$\Delta v = K_v\left(\frac{1}{n}\sum_{i=1}^{n} Q_{i(p.u)} - 1\right). \tag{26}$$

For Equation (26), $K_v = n_1 Q_1^*$. Create a state space model of reactive power per unit value similar to (15) as follows:

$$\begin{cases} \dot{Q}_{i(p.u)}(t) = u_{vi}(t) \\ y_{vi}(t) = Q_{i(p.u)}(t) \end{cases} \tag{27}$$

Build a distributed event trigger controller by:

$$u_{vi}(t) = \gamma \sum_{j\in N_i} a_{ij}\left(\hat{Q}_{j(p.u)}(t) - \hat{Q}_{i(p.u)}(t)\right) \tag{28}$$

$Q_{i(p.u)}(t)$ is substituted into Equation (26) to obtain the voltage compensation amount at the time of DGi:

$$\Delta v_i = K_v\left(Q_{i(p.u)}(t) - 1\right) \tag{29}$$

Based on Equation (21), define the measurement error of state variable $Q_{i(p.u)}$ as:

$$e_{vi}(t) = Q_{i(p.u)}\left(t_k^{vi}\right) - Q_{i(p.u)}(t) \tag{30}$$

Trigger function for each DGi is:

$$f_i(t, e_{vi}(t)) = \|e_{vi}(t)\| - c_v e^{-\alpha_v t} \tag{31}$$

Likewise, $C_v > 0$ and $\alpha$ are normal numbers to be determined. Event trigger time $t_k^{vi}$ is defined as:

$$t_k^{vi} = \inf\left\{t > t_{k-1}^{vi} \middle| f_i(t, e_{vi}(t)) = 0\right\}. \tag{32}$$

The distributed controller satisfying Theorem 1, (27), and (28) guarantees the global stability of a dynamic system with the reactive power per unit. Combining (2), (27), (29), and (30) yields the voltage setting value:

$$v_{ni}(t) = \int [v_i(t) + \Delta v_i(t)]dt. \tag{33}$$

Figure 3 shows the secondary control block diagram. When the DG event is triggered, only the state value can update the DG data holder. Similarly, DGi can only pass its state value to the data holder of the neighboring DG when the event is triggered. As long as the DGi does not generate event trigger control, the data of a corresponding DGi data holder is the states' quantity at the last trigger point. So, the number of communications between

DGs is greatly reduced, the communication pressure on the system is reduced, and the plan becomes more reliable and meaningful.

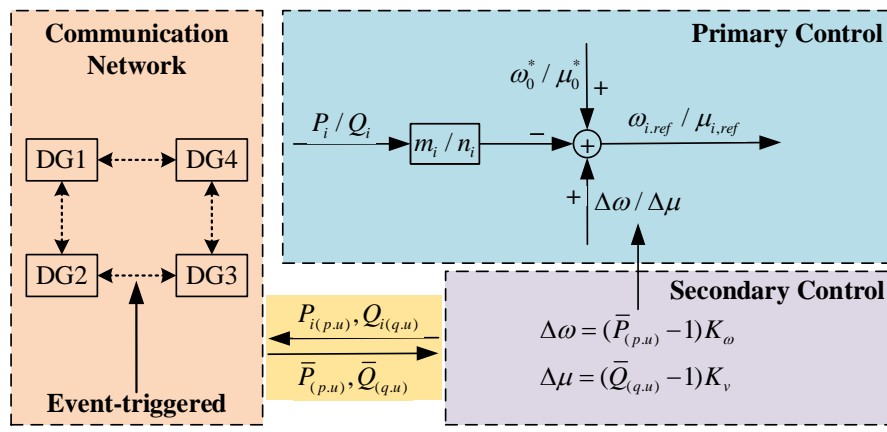

**Figure 3.** Secondary control block diagram.

## 4. Stability Analysis

In this section, we demonstrate the global stability of the above control strategy using Lyapunov's stability theory. We analyzed the lower bound of the event-triggered time interval and show that the event triggering control strategy cannot have an infinite triggering moment within a limited time, i.e., it does not produce Zeno behavior.

### 4.1. Proof of Theorem 1

In order to facilitate writing, abbreviate (15), (19), (21) and (22) as follows:

$$\begin{cases} \dot{x}_i(t) = u_i(t) \\ y_i(t) = x_i(t) \end{cases} \tag{34}$$

$$u_i(t) = \gamma \sum_{j \in N_i} a_{ij} \left( x_i\left(t_k^i\right) - x_j\left(t_k^i\right) \right) \tag{35}$$

$$e_i(t) = x_i\left(t_k^i\right) - x_i(t) \tag{36}$$

$$f_i(t, e_i(t)) = \|e_i(t)\| - ce^{-\alpha t}. \tag{37}$$

By substituting (36) into (35), we get:

$$u_i(t) = \gamma \sum_{j \in N_i} a_{ij}\left((x_i\left(t_k^i\right) - x_i(t)) - \left(x_j\left(t_k^j\right) - x_j(t)\right)\right) = \gamma \sum_{j \in N_i} a_{ij}\left(e_i(t) - e_j(t) + x_i(t) - x_j(t)\right) \tag{38}$$

where it takes stacking abbreviation (38), $E = [e_1(t), \ldots, e_n(t)]^T$, and $X = [x_1(t), \ldots, x_n(t)]^T$ and reduces it to:

$$u(t) = -\gamma L(E + X). \tag{39}$$

The following is the Lyapunov function:

$$V = \frac{1}{2}x(t)^T x(t) \tag{40}$$

In (40), the time reciprocal is:

$$\dot{V} = X^T L\dot{X} = -X^T \gamma L(E + X) = -X^T \gamma LX - X^T \gamma LE. \tag{41}$$

According to Lyapunov's definition, Equation (41) can be written as follows:

$$\dot{V} = -\sum_i X_i^2 - \sum_i \sum_{j \in N_i} X_i(E_i - E_j) = -\sum_i X_i^2 - \sum_i |N_1| X_i E_i + \sum_i \sum_{j \in N_i} X_i E_j \tag{42}$$

Consider the inequality $|xy| \leq (\lambda/2)x^2 + (1/2\lambda)y^2$, then:

$$\dot{V} \leq -\lambda_m \gamma \|X\|^2 + \frac{1}{2}\lambda_m \gamma \|X\|^2 + \frac{\gamma^2 \lambda_n^2}{2\lambda_m}\|E\|^2 = -\frac{1}{2}\left[\lambda_m \gamma \|X\|^2 - \frac{(\gamma \lambda_n)^2}{\lambda_m}\|E\|^2\right] \quad (43)$$

$\lambda_m$ and $\lambda_n$ are the minimum and maximum of the eigenvalues of the matrix $L$, respectively.

Considering Equation (37), Equation (43) and the triggering process, when they are satisfied:

$$ce^{-\alpha t} > \frac{\lambda_m^2 \|X\|^2}{(\gamma \lambda_n)^2} \quad (44)$$

where we get:

$$\dot{V} \leq 0 \quad (45)$$

In conclusion, the system is globally stable.

*4.2. Boundaries of the Event-Triggered Time Interval*

**Theorem 2.** *Assuming that G is connected, $0 < \tau < 1$, and the update interval between each agent is within a time period [9,10]. In the case of Equation (41) satisfying $\dot{V} \leq 0$, the following can be obtained:*

$$\dot{V} \leq -\|LX\|^2 + \|LX\|\|L\|\|E\| \quad (46)$$

The required $E$ satisfies:

$$\|E\| \leq \tau \frac{\|LX\|}{\|L\|} \quad (47)$$

At $t = 1$, we can get $\tau < 1$ and $\|LX\| \neq 0$ when $\dot{V} \leq (\tau - 1)\|LX\|^2$ is negative.

For $t \in [t_k, t_{k+1})$, $\dot{x}(t) = -L(x(t) + e(t))$ can be rewritten as $x(t) = -Lx(t_k)(t - t_k) + x(t_k)$. Equation (47) can be expressed as:

$$\|Lx(t_k)\|(t - t_k) \leq (\tau/\|L\|)\|(-(t - t_k)L + I)Lx(t_k)\| \quad (48)$$

Therefore, the upper bound of the next execution time $t_{k+1}$ is obtained by solving Equation (48):

$$\|Lx(t_k)\|(t - t_k) = (\tau/\|L\|)\|(-(t - t_k)L + I)Lx(t_k)\| \quad (49)$$

This further simplifies to:

$$\left(\|Lx(t_k)\|^2\|L\|^2 - \tau^2\|L^2 x(t_k)\|^2\right)(t - t_k)^2 + 2\tau^2(Lx(t_k))^T LLx(t_k)(t - t_k) - \tau^2\|Lx(t_k)\|^2 = 0 \quad (50)$$

Noted:

$$\left(\|Lx(t_k)\|^2\|L\|^2 - \tau^2\|L^2 x(t_k)\|^2\right) > \left(1 - \tau^2\right)\|Lx(t_k)\|^2\|L\|^2 > 0 \quad (51)$$

$\Delta > 0$, among them;

$$\Delta = 4\tau^4\|Lx(t_k)^T LLx(t_k)\|^2 + 4\tau^2\|L^2 x(t_k)\|^2\left(\|Lx(t_k)\|^2\|L\|^2 - \tau^2\|L^2 x(t_k)\|^2\right) \quad (52)$$

where it draws an upper bound:

$$t = t_k + \frac{-2\tau^2(Lx(t_k))^T LLx(t_k) + \sqrt{\Delta}}{2(\|Lx(t_k)\|^2\|L\|^2 - \tau^2\|L^2 x(t_k)\|^2)} \quad (53)$$

As a result, the next update time boundary is:

$$t_{k+1} - t_k \leq \frac{-2\tau^2 (Lx(t_k))^T LLx(t_k) + \sqrt{\Delta}}{2(\|Lx(t_k)\|^2 \|L\|^2 - \tau^2 \|L^2 x(t_k)\|^2)} \tag{54}$$

In summary, Theorem 2 has been proven.

## 5. Simulation Results

To verify the efficacy of the proposed distributed event-triggered control, the 220 V/50 Hz islanded microgrid test system in Figure 4 was developed in the MATLAB/Simulink simulation platform. The specific system parameters are listed in Table 1.

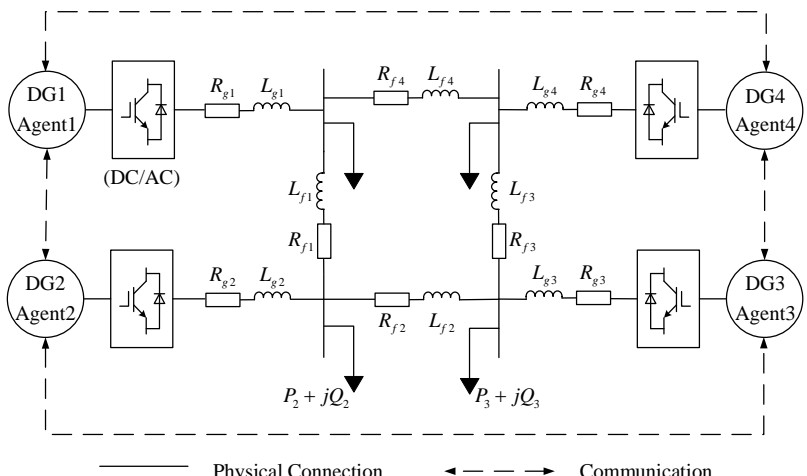

**Figure 4.** Islanded microgrid test system.

**Table 1.** Parameters of the islanded microgrid test system.

|  | **DGs** |  | **DG1** | **DG2** | **DG3** | **DG4** |
|---|---|---|---|---|---|---|
|  | $R_f/\Omega$ |  | 0.01 | 0.01 | 0.01 | 0.01 |
|  | $L_f/mH$ |  | 5 | 5 | 5 | 5 |
|  | $C_f/uF$ |  | 20 | 20 | 20 | 20 |
|  | m |  | $1 \times 10^{-4}$ | $0.5 \times 10^{-4}$ | $1.33 \times 10^{-4}$ | $0.67 \times 10^{-4}$ |
|  | n |  | $2 \times 10^{-3}$ | $2 \times 10^{-3}$ | $2 \times 10^{-3}$ | $2 \times 10^{-3}$ |
|  | $R_g/\Omega$ |  | 0.01 | 0.01 | 0.01 | 0.01 |
|  | $L_g/mH$ |  | 20 | 20 | 20 | 20 |
|  | P/kW |  | 11 | 7 | 3 | 15 |
|  | Q/kvar |  | 1 | 1 | 1 | 1 |
| $C_\omega$ | | $C_v$ | 1.1 | 1.1 | 1.1 | 1.1 |
| $\alpha_\omega$ | | $\alpha_v$ | 1 | 1 | 1 | 1 |

According to [11,12], the islanded microgrid test system shown in Figure 4 was equivalent to MAS. The communication topology used by this control system was connected and included a directed spanning tree.

### 5.1. Performance of Secondary Control Simulation by the Distributed Event-Triggered Method

The simulation time of this system was 0–1.5 s. From the beginning, the system control was activated and directly connected to the secondary control. The simulation results are shown in Figure 5. Due to the characteristics of the primary droop control, all voltages and frequencies deviated from the nominal values when the simulation system was initially activated. Under the action of the event-triggered controller, both frequency and voltage can return gradually to the nominal value at a time of about $t = 0.3$ s, thus $\omega_{ref} = 50$ Hz and $V_{ref} = 311$ V. In a stable state, accurate active power sharing can also be guaranteed.

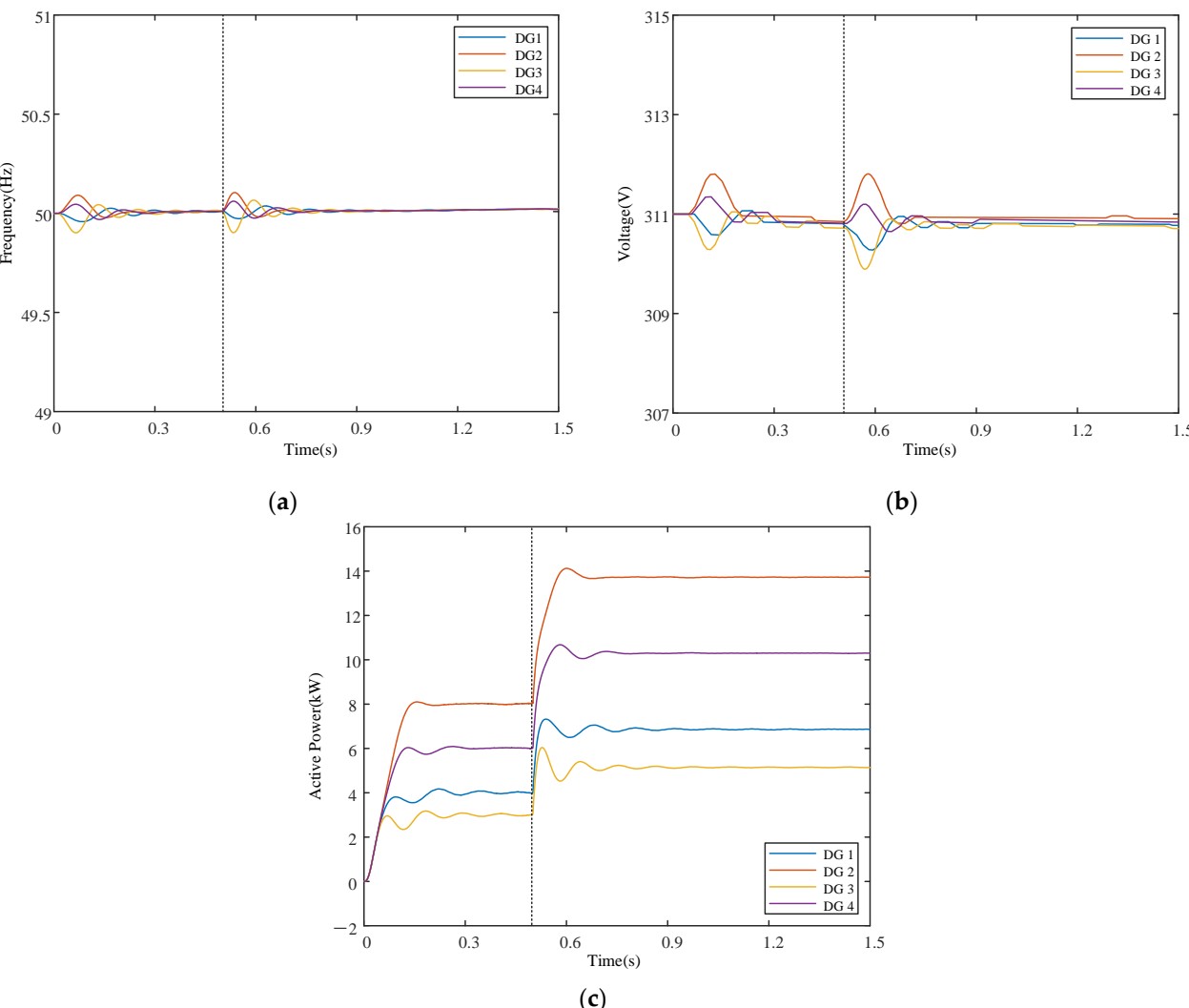

**Figure 5.** (**a**) Frequency output waveform; (**b**) voltage output waveform; (**c**) active power output waveform.

As shown in Figure 5c, in 0–0.5 s, the active power for DG1–GD4 was 21 kW load start and the active power distribution ratio was 4:8:3:6. After 0.5 s, the system active load increased by 15 kW, and each DG still bears the load based on the initial active power distribution ratio. When $t = 0.8$ s, the frequency and voltage can gradually return to the nominal value and when $t = 0.7$ s, the DG output continues to be distributed according to the original ratio, which is 6.857 kW, 13.714 kW, 5.143 kW, and 10.286 kW. Therefore, the output of each DG is more realistic within a given error range, which proves that the secondary control strategy is effective.

To demonstrate the superiority of the proposed consistent event trigger control strategy, Figures 6 and 7 show the measurement error of the frequency and voltage of each DG from the start of the controller up until the end of the simulation, a second norm $e_i(t)$, and trigger threshold $ce^{-\alpha t}$. The trigger point is formed when $e_i(t)$ equals $ce^{-\alpha t}$, that is, at the intersection of the blue line and the red dashed line in the figure.

As shown in Figures 6 and 7, the schematic diagram of each DG event trigger time was calculated, as shown in Figure 8.

Figure 8 illustrates that in 0–0.5 s, when the system was activated for the first time, the controller started triggering frequently. At about $t = 0.3$ s, the frequency and voltage gradually returned to the nominal value and the number of triggers was greatly reduced. At $t = 0.5$ s, the load suddenly increased and the controller began to trigger more intensively. Until approximately $t = 0.8$ s, when the frequency and voltage gradually returned to

their nominal values, the triggering situation decreased. A trigger situation can be seen, according to the simulation output diagram in Figure 5. Specifically, this paper shows how a distributed event-triggered control strategy can be applied to the islanded microgrid.

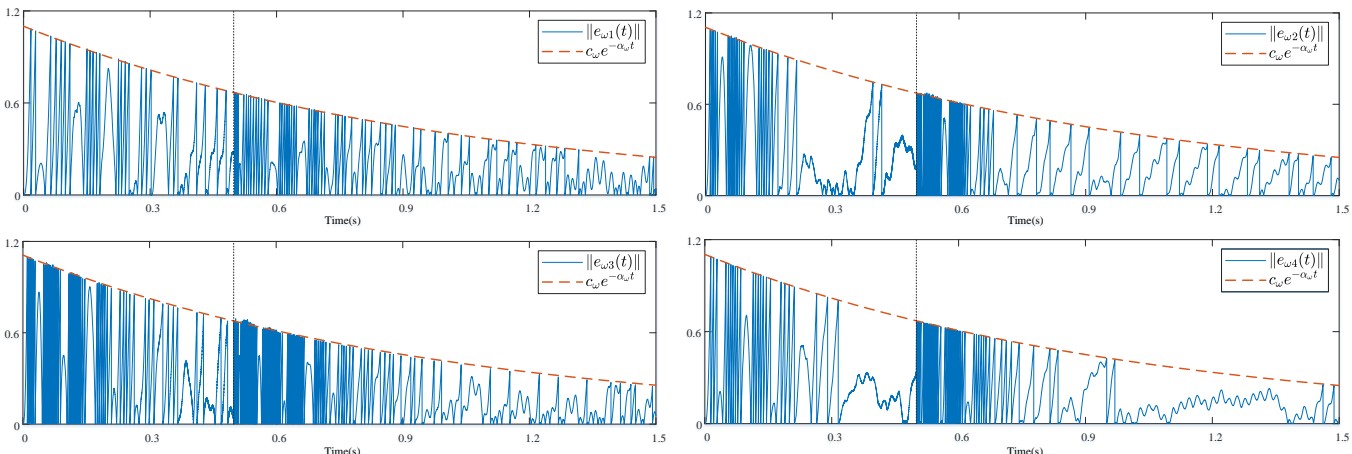

**Figure 6.** The frequency trigger state diagram for each DG.

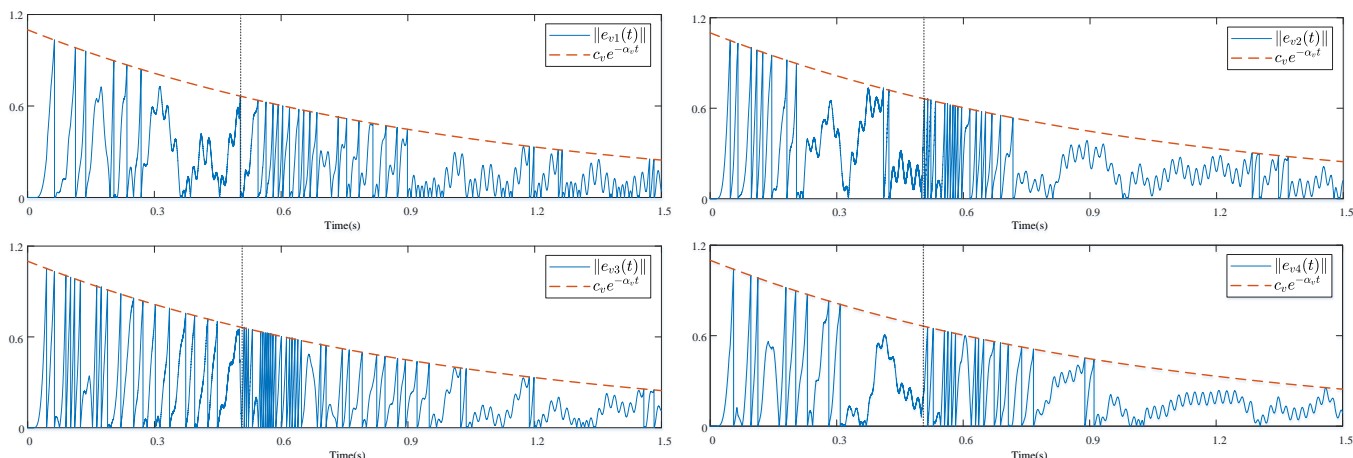

**Figure 7.** Voltage trigger state diagrams for each DG.

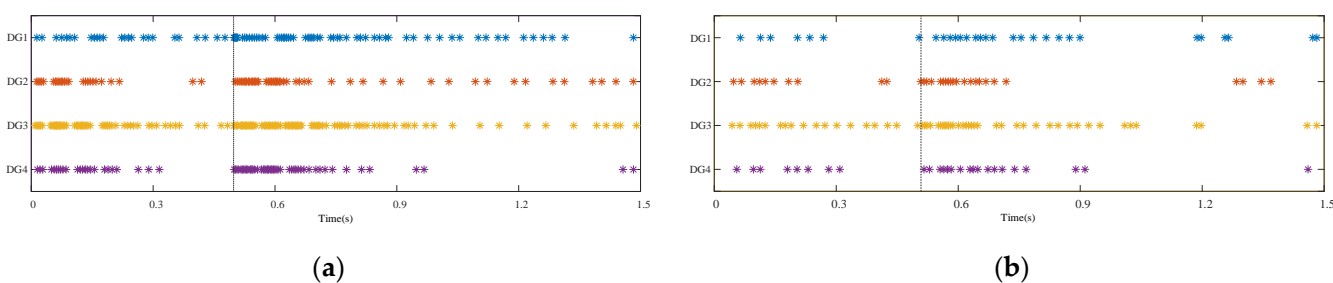

(**a**)                                                     (**b**)

**Figure 8.** (**a**) Schematic diagram of the frequency trigger time for each DG; (**b**) schematic diagram of voltage trigger times for each DG.

### 5.2. Compared to Traditional Distributed Control Methods

Generally, islanded microgrid control uses periodic feedback [17–19]. The simulation results for the same environment, compared with the traditional distributed control method, are shown in Figure 9. Similar to that, in 0–0.5 s the initial islanded microgrid was activated. After 0.5 s, the load increased suddenly. The relevant parameters of each DG remained the same. It is evident that without the event-triggered mechanism, the times when the frequency, voltage, and active power reached stability were all advanced, which were 0.1 s,

0.2 s, and 0.1 s respectively. In comparison with the control situation with the event trigger mechanism, the convergence speed was faster, the stabilization time was extended, and the stabilization effect was better. Following a sudden increase in load, at $t = 0.5$ s, the relevant parameters of each DG varied greatly. The traditional distributed control used the periodic state feedback and the controller was always in the trigger state. Then it had feedback calculations for each run.

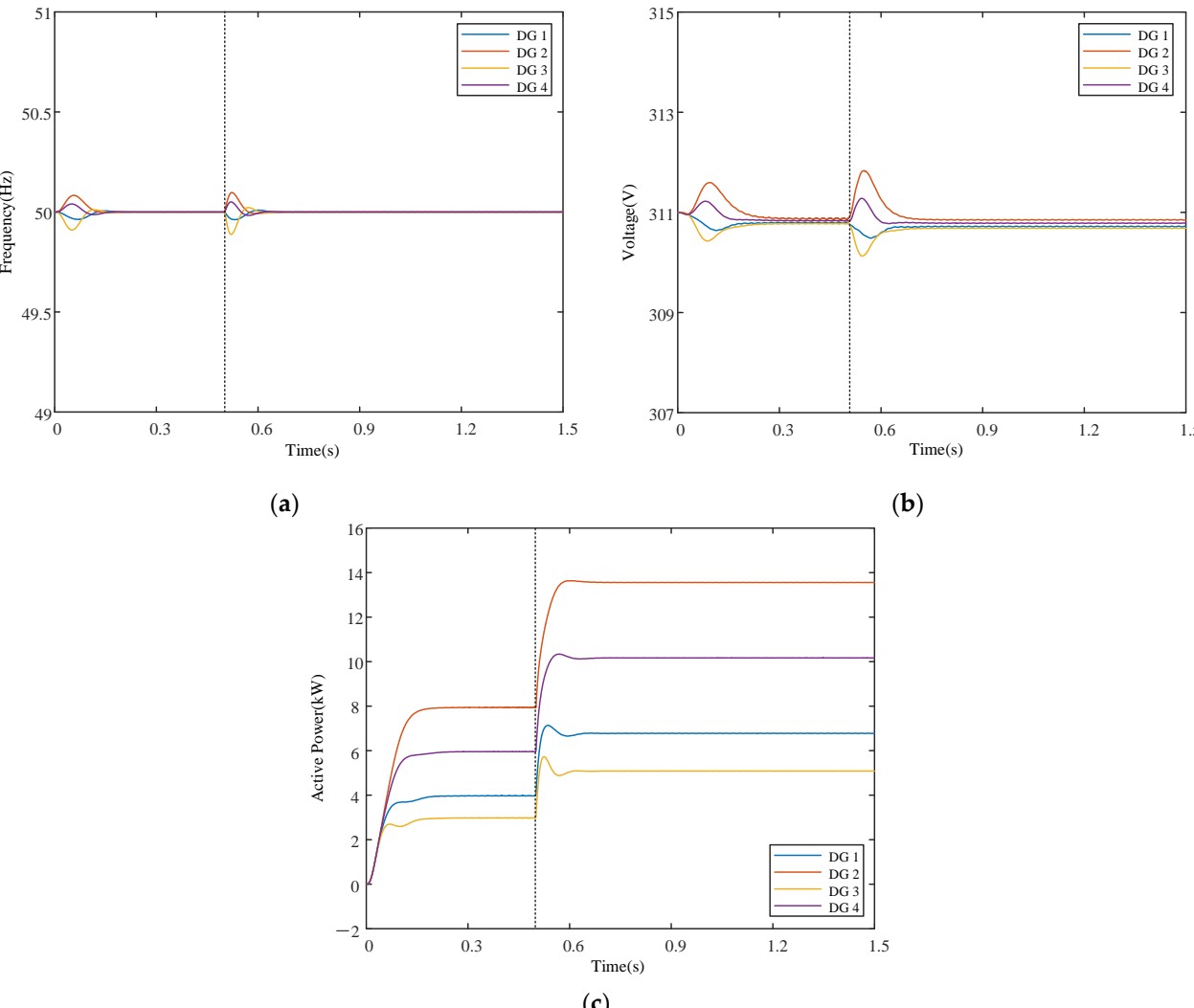

**Figure 9.** (**a**) Frequency output waveform; (**b**) voltage output waveform; (**c**) active power output waveform.

With DG1 as an example, this paper compares the proposed event-triggered control with the traditional distributed control using the same control gain. The simulation results are shown in Figure 10a–c. It is apparent that the proposed event-triggered method can also have a stable time as the traditional control method. This further demonstrates the effectiveness of the proposed control strategy. However, the event-triggered method had more oscillations. As the state is updated when an event is triggered, the sawtooth wave can be observed in an event-triggered manner.

To illustrate how the proposed control strategy communicates and to compare its communication burden with the traditional method, Figure 10d shows the event trigger time of the distributed DG frequency and voltage control for DG1. For the traditional communication method, the time interval considered was 5 ms. Table 2 lists the trigger results of each DG at a simulation time of 0–1.5 s. Combining Figure 10d and Table 2, it can be seen intuitively that the distributed event-triggered control strategy described here was capable of meeting the operating conditions for the islanded microgrid and restoring the

voltage and frequency to standard, while at the same time reducing the communications between the DGs.

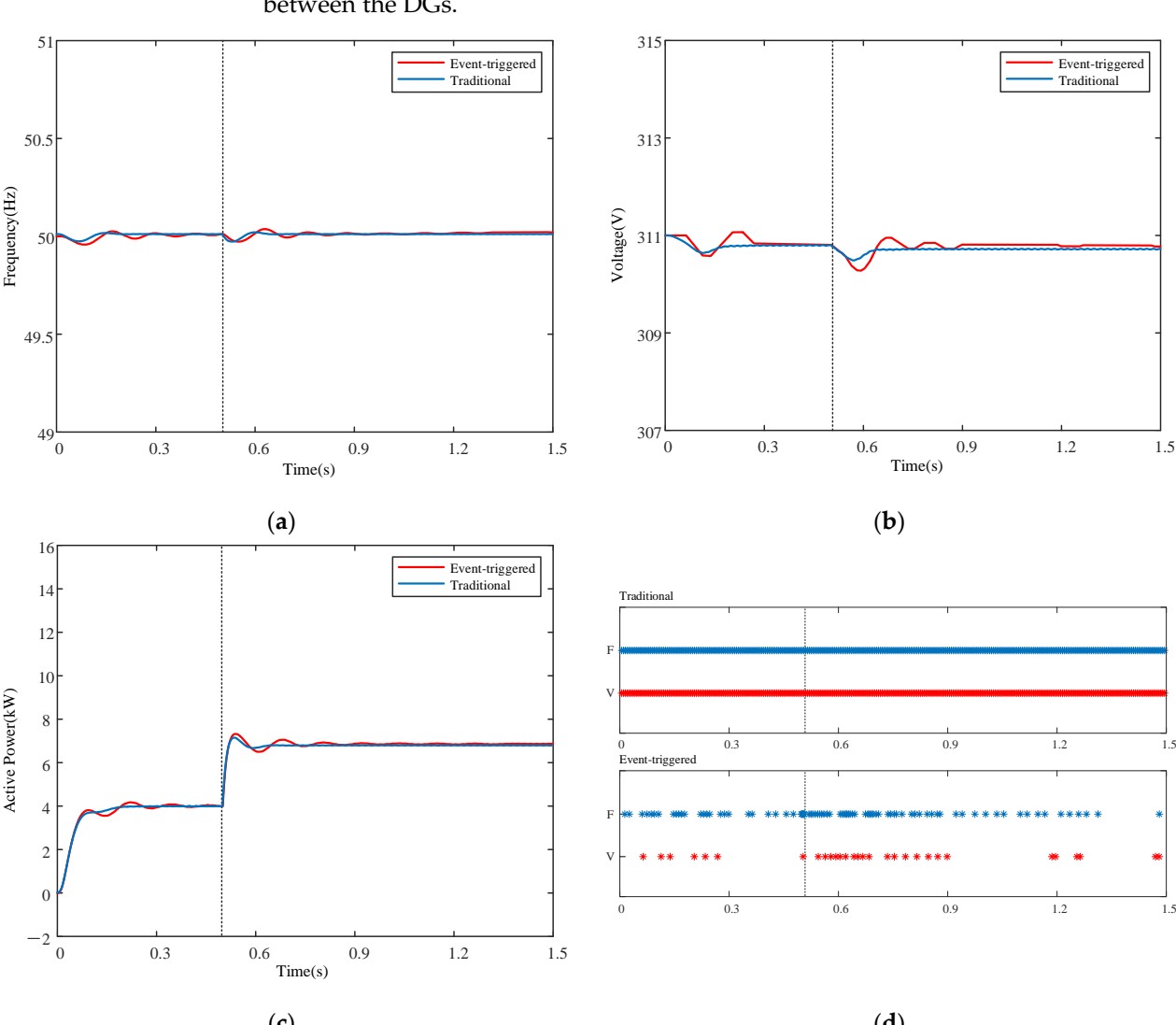

**Figure 10.** (**a**) Frequency output waveform of DG1; (**b**) voltage output waveform of DG1; (**c**) active power output waveform. (**d**) Trigger event instants for DG1.

**Table 2.** Communication result.

| DGs | | Number of Communications | | Average Interval between Two Consecutive Communications | |
|---|---|---|---|---|---|
| | | Traditional | Event-triggered | Traditional | Event-triggered |
| DG1 | Frequency | 300 | 86 | 5 ms | 18 ms |
| | Voltage | 300 | 30 | 5 ms | 50 ms |
| DG2 | Frequency | 300 | 79 | 5 ms | 19 ms |
| | Voltage | 300 | 32 | 5 ms | 47 ms |
| DG3 | Frequency | 300 | 138 | 5 ms | 11 ms |
| | Voltage | 300 | 56 | 5 ms | 27 ms |
| DG4 | Frequency | 300 | 73 | 5 ms | 21 ms |
| | Voltage | 300 | 26 | 5 ms | 58 ms |

## 6. Conclusions

Based on the primary control, this paper applied the distributed event-triggered control to the secondary recovery control of the islanded microgrid. The use of distributed event-triggered control not only realized the frequency and voltage recovery of the is-

landed microgrid, but also satisfied the sharing of active power according to the original proportion. Compared with the traditional distributed power grid control, the control method adopted in this paper greatly reduced the communication frequency between DGs and avoided continuous communication between DGs. In the actual operation of the microgrid, the distributed event-triggered control method reduced the requirements on the communication network and had more practical significance.

This paper only considered the frequency and voltage recovery and the distribution of active power in accordance with the original proportions. It is necessary to further analyze the problem of active power distribution. In addition, the packet loss and time delay of the communication network are also factors that affect the control of event triggers. At the same time, research on how to further improve the response speed of the proposed secondary controller still needs further research.

**Author Contributions:** X.W. and Y.T. conceived and designed the study; X.W. and Y.T. performed the study; X.W., J.W., and X.D. reviewed and edited the manuscript; and X.W., Y.T. and H.T. wrote the paper. All authors read and agreed to the published version of the manuscript.

**Funding:** This research was funded by National Natural Science Foundation of China, grant number 61563034.

**Data Availability Statement:** The data that support the findings of this study are included within the article.

**Conflicts of Interest:** The authors declare no conflict of interest regarding the publication of this paper.

## Nomenclature

| | |
|---|---|
| $\omega_i, \omega_{ni}, \omega_{ref}$ | Output frequency, its set-value and reference value of DGi. |
| $v_i, v_{ni}, v_{ref}$ | Output voltage, its set-value and reference value of DGi. |
| $m_i, n_i$ | Frequency and voltage droop coefficients. |
| $p_i, P_i, P_i^*$ | Output, filtered and rated active power. |
| $q_i, Q_i, Q_i^*$ | Output, filtered and rated reactive power. |
| $P_{i(p.u)}, Q_{i(p.u)}$ | Activated power per unit value and reactive power per unit value. |
| $\omega_c$ | Cutoff frequency of low-pass filter. |
| $i_{di}, i_{qi}$ | Output frequency in the $dq$ coordinate of DGi. |
| $v_{di}, v_{qi}$ | Output voltage in the $dq$ coordinate of DGi. |
| $\Delta\omega_i$ | Frequency compensation average value of DGi. |
| $u_{\omega i}(t), u_{vi}(t)$ | Frequency and voltage control inputs of DGi. |
| $y_{\omega i}(t), y_{vi}(t)$ | Frequency and voltage control outputs of DGi. |
| $t_k^{\omega i}, t_k^{vi}$ | Event-triggered time of the frequency and voltage control for DGi. |
| $e_{\omega i}(t), e_{vi}(t)$ | Event-triggered state measurement error of the frequency and voltage for DGi. |

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
