# Peer review of "Distributed Event-Triggered Secondary Recovery Control for Islanded Microgrids"

_electronics, doi:10.3390/electronics10151749_

Round 1

Reviewer 1 Report

This work proposes a distributed event-triggered secondary recovery control strategy for island microgrids. The work is generally well-written: the methodology and algorithms have been clearly presented, and the results have shown the effectiveness of the proposed strategies.

  1. The authors are suggested to carefully proofread the work since it contains some grammatical problems and typos. E.g.
    1. Line 89, GD4 -> DG4
    2. Line 155, island microgrid -> islanded microgrid
    3. Line 172, state space -> state-space
    4. Line 234, the sentence is grammatically wrong.
    5. Line 235, Lyapulov -> Lyapunov
    6. Line 272, Simlink-> Simulink
    7. Line 283, Figures 5 -> Figure 5
    8. Use ‘equation’ instead of ‘formula’
  2. The significance of the work has to be emphasized more clearly in the introduction.
  3. There is a lack of comparative study to show the superiority of the proposed algorithm.

Reviewer 2 Report

This manuscript discusses an important and recent issue, it adds to knowledge and contains valuable information, the authors organize the manuscript well. The authors should accurately address the below comments.

  • Abstract: The authors indicated the purpose of this research at the beginning of the abstract, however, that it should be written better to be clear to readers and researchers. Also, it is preferable to add some numerical findings from the analyzes at the end of the abstract.
  • Keywords: We suggest that the authors should replace keywords such as “islanded microgrid” and “event-triggered” because these keywords are already found in the review manuscript title. It is better that they replace them with other keywords to increase the reach of the manuscript.
  • Some abbreviations require to define before use such as MAS (page2), IGBT (page3) … etc.
  • Introduction Section: It contains valuable information. However, the names of researchers should be added in addition to the reference number (page1-line34 and line38 … etc.). The authors referred to the phrase “In this article", which one the authors meant [4], [5] or [8] (page1-lines40-41).
  • Results: Discussing the results requires more clarification, especially with regard to comparison with existing research schemes.
  • Conclusion Section: It did not describe the conclusion of this study clearly.
  • Figures and Tables: All Figures and Tables are drawn with high resolution.
  • English writing: This manuscript requires minor proofreading (typo and grammar). The authors should carefully scrutinize the entirety of the manuscript.
  • List of References: The references should follow Electronics-MDPI style. The references are related to the manuscript topic but are not enough. Some search names in the reference list begin an uppercase letter for each word (such as [2], [3], [4] ... etc.) and others use only an uppercase letter in the first word (such as [1], [5] … etc.), authors should standardize style. Journal names should be in italics such as [9].  Some of the references do not contain enough information such as [9], [13] … etc. The authors should carefully check list of references to remove all problems.

Reviewer 3 Report

The authors should address the comments below:

1- The text requires thorough proofreading. There are several grammatical errors to be corrected. Some examples are as follows: Line 13, Line 48, Line 86-87.

2- The conclusion is too general. Would you please rewrite the conclusion specifically using the outcome of your work? 

3- Line 44, the reference [10] introduces a decentralized controller, not a centralized one. 

4- In referring to your references, use the number in the bracket without using the term "Reference" or "Literature", e.g., Lines 51, 68, 70, 71. 

5- Introduce abbreviations at their first appearance in the text. For example, MAS in Line 86. 

6- The variables and parameters should be appropriately introduced. What are delta omega and delta v in (11) and (12)? or Pstar in Line 168? I would suggest adding a nomenclature to the manuscript. 

7- It seems a correction is required in (13); please double-check the correctness of delta in the last term. 

Round 2

Reviewer 1 Report

Thanks the authors. The manuscript has been improved based on my previous comments and I do not have further comments.